# LUNA EMG as a Marker of Adherence to Prehabilitation Programs and Its Effect on Postoperative Outcomes among Patients Undergoing Cytoreductive Surgery for Ovarian Cancer and Suspected Ovarian Tumors

**DOI:** 10.3390/cancers16142493

**Published:** 2024-07-09

**Authors:** Marcin Adam Zębalski, Krzysztof Parysek, Aleksandra Krzywon, Krzysztof Nowosielski

**Affiliations:** 1Department of Gynecology, Obstetrics and Gynecological Oncology, University Clinical Center of the Medical University of Silesia, 40-752 Katowice, Poland; marcin.zebalski@sum.edu.pl; 2Department of Movement Rehabilitation and Physical Therapy, University Clinical Center of the Medical University of Silesia, 40-752 Katowice, Poland; parys3k@gmail.com; 3Department of Biostatistics and Bioinformatics, Maria Sklodowska-Curie National Research Institute of Oncology, Gliwice Branch, 44-102 Gliwice, Poland; aleksandra.krzywon@gliwice.nio.gov.pl

**Keywords:** prehabilitation, LUNA EMG, ERAS, ovarian cancer

## Abstract

**Simple Summary:**

Prehabilitation is a multimodal intervention including preoperative exercises, a high-protein diet, psychological support, smoking/alcohol cessation, and the optimization of preoperative laboratory results. The effectiveness of prehabilitation is different for each patient and depends on many factors. Ensuring reliable compliance with prehabilitation recommendations and active patient involvement are pivotal. To accurately assess patient adherence to prehabilitation guidelines, it is crucial to employ innovative assessment tools. To the best of our knowledge, this is the first study utilizing a LUNA EMG device as a marker of prehabilitation compliance. In this study, we implemented a prehabilitation program in a group of patients with suspected ovarian cancer and compared the results with those of the control group in which only the ERAS protocol was used. We observed an improvement in muscle strength and tension during the prehabilitation program and found an association between prehabilitation using this device and fewer complications and shorter hospital stays compared to the control group.

**Abstract:**

Background: Prehabilitation is a novel strategy in preoperative management. The aim of this study was to investigate the effect of prehabilitation programs on peri- and postoperative outcomes and to verify if LUNA EMG has the capacity to monitor compliance with prehabilitation programs. Methods: A total of seventy patients with suspected ovarian cancer were recruited between April 2021 and September 2022 and were divided into a prehabilitation group (36 patients) or a control group (34 patients). A LUNA EMG device was utilized to monitor muscle strength and tension. Results: Within the prehabilitation group, we observed a significant increase in the 6-Minute Walk Test distance by 17 m (median, IQR: 0–42.5, *p* < 0.001) and a significant increase in muscle strength measured with LUNA EMG. In comparison to the control group, the prehabilitation group showed fewer complications according to the Clavien–Dindo classification (47.2% vs. 20.6%, *p* = 0.02) and shorter postoperative hospital stays (median 5.0 days [IQR: 4.0–6.2] vs. 7.0 days [IQR: 6.0–10.0], *p* < 0.001). Conclusion: Prehabilitation has a positive effect on physical capacity and muscle strength and is associated with a reduction in the number of complications after surgery. LUNA EMG can be a useful tool for monitoring patients’ adherence to prehabilitation programs.

## 1. Introduction

Ovarian cancer is typically diagnosed in advanced stages, spreading within the abdominal cavity, and usually requires extensive primary debulking surgery, followed by adjuvant chemotherapy. However, these surgeries can be complex, often involving bowel and multiorgan resections and prolonged duration, leading to a high risk of postoperative complications. To maximize treatment effectiveness, the complete gross resection (R0) of all visible tumor foci is essential. Adjuvant chemotherapy administered promptly post-surgery has been shown to enhance outcomes, as delayed initiation can adversely impact overall survival. Moreover, minimizing surgical complications is crucial for optimizing the overall treatment effect [1,2,3].

To minimize the risk of post-procedure complications, a prehabilitation program was proposed. Prehabilitation, a multimodal approach increasingly utilized in gynecological cancer surgery, focuses on preparing patients for surgery during the preoperative period. These programs incorporate preoperative exercises, high-protein nutrition, psychological support, smoking cessation, and the application of Enhanced Recovery After Surgery (ERAS) protocols [4,5,6,7]. Prehabilitation comprises a holistic approach aimed at shortening recovery periods, reducing complications, and enhancing the overall quality of life. Studies have shown that patients undergoing preoperative multimodal interventions experience fewer complications and shorter hospitalization stays compared to those without prehabilitation [8,9,10,11,12,13,14].

In recent years, there has been a growing interest in prehabilitation, yet there remains a notable shortage of randomized controlled trials, particularly in gynecology. Many existing studies offer prehabilitation programs with recommended physical activities discussed in a generalized manner, lacking detailed specifics. Moreover, there are no objective methods to assess the effectiveness of prehabilitation interventions.

The aim of the study is to assess the impact of multimodal preparation in the prehabilitation protocol on the incidence of postoperative complications and duration of hospitalization after surgery among women scheduled for cytoreductive surgery due to advanced-stage ovarian cancer. The second aim is to evaluate if LUNA EMG might serve as a useful tool to assess patient compliance with a prehabilitation protocol.

To achieve this, we opted to employ a LUNA EMG device for measuring muscle tension. This innovative neurorehabilitation device utilizes bioelectric signals from the patient’s muscles to measure minimum, maximum, and average muscle tension and tone. By recording the electrical activity of muscle fibers and conducting subsequent analysis, the LUNA EMG device provides an effective and objective means to assess muscle strength and tension [15].

Our study is a pioneer in using the LUNA EMG robot as a marker of compliance with prehabilitation recommendations. This approach aims to shed light on the impact of prehabilitation on the length of hospital stay and postoperative complications following gynecological oncology procedures.

## 2. Materials and Methods

### 2.1. Participants

Patients were recruited from 1 April 2021 to 31 September 2022. The patients included in this study were scheduled for surgery at the Clinic of Obstetrics, Gynecology, and Oncologic Gynecology in Katowice, Poland, due to ovarian cancer or suspected ovarian tumors.

In the first step, the patient was referred to a registration officer in the outpatient clinic who assigned the numbers one or two to every second patient. The registration officer had no knowledge of the patients’ medical histories. In the next step, patients assigned the number one were invited by the doctor supervising the prehabilitation program to an introductory visit. The patients had no contact with each other and were not aware of which group they were assigned to by the registration officer. Completely random assignment to both groups by the registration officer allowed for the avoidance of selection bias.

Inclusion criteria comprised willingness to participate in the study, confirmed or suspected ovarian cancer, and performance status classified as Eastern Cooperative Oncology Group (ECOG) < 2. Exclusion criteria included patients with comorbidities that disqualified them from extensive surgery based on the G8 (Geriatric 8) scale and MSK (Memorial Sloan Kettering Cancer Center) protocol [16] and those with neurological or functional limitations contraindicating exercise and fitness assessment, such as body paresis, severe cerebral dementia, or condition following limb amputation. An additional exclusion criterion was the patient’s lack of consent to participate in the study. The advancement of the disease and the inability to perform radical cytoreductive surgery were not exclusion criteria from the study, although these patients were excluded from the analyses for this publication. All patients not suitable for cytoreductive operation were scheduled for laparoscopy or CT-guided (computed tomography-guided) biopsy to obtain tissue samples. If ovarian cancer was confirmed, those patients were referred for neoadjuvant chemotherapy and then reassessed after three cycles.

This was a single-center analytical prospective randomized interventional study.

### 2.2. Study Protocol

The ERAS protocol was implemented for both groups. Patients in the prehabilitation group underwent the prehabilitation program introduced during the initial visit, approximately two to four weeks prior to the scheduled surgery. During this visit, patients were evaluated in terms of physical activity, nutritional status, and symptoms of anxiety and depression. Baseline medical optimization and assessments were conducted, including blood tests to evaluate total protein, albumin, and blood count to detect possible anemia; vitamin D levels; and thyroid, kidney, and liver function. Any abnormalities detected in these tests were addressed, and patients with disturbances were referred for specialist consultation.

Patients were educated about the importance of adhering to a high-protein diet and additional protein supplementation while also being advised against alcohol consumption. Smokers were encouraged to quit, and support resources for smoking cessation were provided.

Furthermore, during the initial visit, patients were instructed on the exercises comprising the prehabilitation program, and they were guided through each exercise to ensure proper understanding and execution.

On the day of hospital admission, prior to surgery, physical activity, nutritional status, and mental health were assessed for both study and control group patients. Patients in the control group underwent evaluation upon admission to the hospital.

Patients from the control group were assessed on the day of admission to the hospital in the same way as patients from the prehabilitation group, and after surgery, their recovery was identical to that of patients from the prehabilitation group, i.e., early mobilization, early oral nutrition, postoperative rehabilitation, and other components of the ERAS protocol.

### 2.3. Tools

To assess physical capacity, patients were asked to perform a 6-Minute Walk Test [6MWT]. Prior to the 6MWT, basic vital parameters like pulse, blood pressure, and oxygen saturation were measured. The physical capacity was evaluated based on the result of 6MWT and maximal oxygen consumption (VO2 max) calculated from the following formula [17]:VO2 max (mL/kg^−1^⋅min^−1^) = 70.161 + (0.023 × 6MWT [m]) − (0.276 × weight [kg]) − (6.79 × sex, where m = 0, f = 1) − (0.193 × resting HR [bpm]) − (0.191 × age [y])

To assess muscle strength and tone among the study participants, the LUNA EMG device was used. Patients in the prehabilitation group underwent EMG analysis twice: first during the introductory visit to the prehabilitation program and then on the day before their scheduled surgery, upon admission to the hospital. Patients in the control group underwent EMG measurement only once, on the day of hospital admission. Muscle tension was assessed in the abdominal rectus muscles of each patient, with measurements taken both at rest and during muscle contraction. The initial measurement was conducted with the patient lying supine (relaxation), followed by a second measurement while the patient was lifted into a seated position without using their hands (contraction).

The assessment of quality of life was performed using the standardized EORTC Quality of Life QLQ-C30 questionnaire [18]. Additionally, to identify patients experiencing symptoms of depression or anxiety, each participant completed the HADS (Hospital Anxiety and Depression Scale) questionnaire [19]. Those scoring 8 or higher were referred to a psychologist for further evaluation.

To determine malnutrition or risk of malnutrition, patients completed the MNA (Mini Nutritional Assessment) and MUST (Malnutrition Universal Screening Tool) questionnaires [20]. A score of 24 or higher on the MNA questionnaire indicated normal nutritional status, while patients scoring two or more points on the MUST questionnaire were referred to a dietitian for further assessment. The G8 scale was also administered to identify elderly patients at risk of frailty, with a score of 14 or lower considered abnormal [21].

### 2.4. Prehabilitation Program

I—First pillar—Physical Activity: Following medical examination, interviews, and assessments such as the 6-Minute Walk Test (6MWT) and LUNA EMG tests, patients were educated on the importance of physical activity in preparing for surgery. They were advised to engage in two types of physical activity: resistance exercises and cardio-type aerobic exercises. Patients were encouraged to exercise regularly, either daily or every other day, and to record their progress while gradually increasing the intensity and duration of their workouts. For safety, patients were instructed to monitor their heart rate during exercise and to perform workouts under the supervision of family members or close relatives.

A set of resistance exercises, tailored to enhance muscle strength, particularly in the abdominal and core muscles, was developed by hospital physiotherapists. During the initial implementation of the prehabilitation program, each patient received personalized instruction from a physiotherapist who demonstrated the exercises and guided the patient through a training cycle. Subsequently, patients had access to instructional videos demonstrating proper exercise techniques on the YouTube platform. Additionally, patients could opt for personal consultations with hospital staff for further guidance during the prehabilitation period.

To support adherence to the program, all patients received an instructional booklet written in simple language with illustrations, detailing the recommended exercises and program elements.

II—Second pillar—Proper Diet and Protein Supplementation: Patients were advised to adhere to a specialized diet regimen as part of their prehabilitation program. Those scoring 0 or 1 point in the MUST nutritional status assessment received specific dietary recommendations to follow at home. Patients with two or more MUST points were referred to a dietitian for further guidance.

The recommended daily protein intake was set at 1.5–2.0 g of protein per kilogram of body weight. Dietary guidelines emphasized the importance of limiting carbohydrates, particularly simple carbohydrates; eliminating processed foods; and increasing the consumption of vegetables rich in vitamins.

In addition to dietary changes, patients were encouraged to incorporate protein supplements into their daily regimen. We particularly recommended supplements containing ingredients known for their immune-supporting, anti-inflammatory, and immunomodulatory properties, such as omega-3 fatty acids, beta-glucan, eicosapentaenoic acid (EPA), or docosahexaenoic acid (DHA).

III—Third pillar—Eliminating Risky Behaviors and Reducing Stimulants Use: Patients were instructed on the importance of limiting alcohol consumption and encouraged to quit smoking. They were educated about the adverse effects of stimulant use during the preoperative period. Long-term smokers struggling to quit were advised to reduce cigarette intake or switch to nicotine patches.

IV—Fourth pillar—Psychological Support in the Preoperative Period: Patients scoring 8 or more points on the HADS questionnaire were referred to a psychologist for further evaluation and support. Additionally, patients who wished to receive psychological counseling were given the opportunity to do so during the pretreatment period. Continuous communication with a supervising doctor ensured patient trust, encouraged active participation in the prehabilitation plan, and facilitated the prompt resolution of any issues that arose.

V—Fifth pillar—Optimization of Laboratory Test Results: During the initial visit, patients underwent laboratory tests to evaluate peripheral blood count, liver function, kidney function, thyroid function, nutritional status, and vitamin D levels. Any deviations from normal values were addressed during the prehabilitation period, with patients referred to specialists as needed to normalize results.

The course of the prehabilitation program is shown in Figure 1.

### 2.5. Enhanced Recovery after Surgery (ERAS) Program

The ERAS protocol encompassed several key components aimed at optimizing patient recovery, namely the avoidance of bowel preparation, nasogastric tubes, and drains; the shortening of the fasting period; carboloading, i.e., the administration of oral carbohydrates 2 h prior to surgery; the early initiation of oral nutrition with a focus on high protein intake; the utilization of multimodal analgesia techniques; venous thromboembolism prophylaxis; and the early initiation of physical activity.

### 2.6. Statistical Analysis

Categorical variables were summarized as frequencies and percentages, while continuous data were summarized as median values with interquartile ranges (IQR, 25% to 75%). Differences between the two groups were assessed using the Wilcoxon rank-sum test for continuous variables and Fisher’s exact test for categorical variables. Changes in measured parameters between two time points were assessed using the one-sample Wilcoxon signed-rank test. To examine the correlations between variables, Spearman correlation coefficients were calculated. Correlations were classified as follows: 0.0 ≤ |r| < 0.1 (negligible correlation), 0.1 ≤ |r| ≤ 0.39 (weak correlation), 0.4 ≤ |r| ≤ 0.69 (moderate correlation), 0.7 ≤ |r| ≤ 0.89 (strong correlation), and 0.9 ≤ r ≤ 1 (very strong correlation) [22]. Odds ratios (ORs) with 95% confidence intervals (CIs) were estimated by univariate and multivariate logistic regression models. Variables with *p*-value < 0.20 in the univariate analysis were included in the multivariate analysis. Statistical significance was defined as a two-sided *p*-value < 0.05. All computational analyses were conducted using the R environment for statistical computing, version 4.0.1 “See Things Now,” released on 6 June 2020, by the R Foundation for Statistical Computing, Vienna, Austria (http://www.r-project.org, accessed on 29 February 2000). After the statistical analysis, a sample size was calculated to verify whether there was adequate statistical power to detect a prehabilitation effect. The sample size was calculated to be 66 patients (33 patients per group), achieving 80% power and a significance level of α = 0.05.

## 3. Results

A total of 70 patients with confirmed or suspected ovarian cancer were enrolled in this study, with 36 patients assigned to the prehabilitation group, where the prehabilitation program was implemented, and 34 patients forming the control group. The average duration of the prehabilitation intervention was 24.5 days (±26.9). The general characteristics of both groups are summarized in Table 1, demonstrating comparability in terms of age, BMI, level of fitness, baseline laboratory results, comorbidities, and social status.

### 3.1. Surgery

All patients underwent laparotomy surgery either for ovarian cancer or suspected ovarian tumors. Ovarian cancer was histopathologically confirmed in 28 (77.8%) patients from the prehabilitation group and 28 (82.4%) patients from the control group. The average operating time was 215.0 ± 108 min in the prehabilitation group and 260.0 ± 93 min in the control group (*p* = 0.08). Patients with advanced disease, for whom cytoreductive surgery was not feasible due to the disease progression observed during laparoscopy, were excluded from analyses for this study. Only patients undergoing radical surgery in laparotomy were included in this study. In the prehabilitation group, 32 patients (89%) underwent primary surgery, 4 patients (11%) underwent interval debulking surgery, and no patients underwent secondary cytoreduction. In the control group, these numbers were 26 patients (76%), 4 patients (12%), and 4 patients, respectively. Patients from the control group and the prehabilitation group were comparable in terms of procedure time, type of surgery (primary, interval, or secondary cytoreduction), degree of radicality, and the Aletti complexity score and peritoneal cancer index (PCI score) (Table 2). The type of operational procedures performed is presented in Table 3. Among the enrolled patients, ovarian cancer was confirmed in 56 (80%) patients. Histopathological characteristics are presented in Table 4.

### 3.2. Results of the 6-Minute Walk Test and Muscle Strength Measured with the LUNA Device

During the prehabilitation period, a significant increase was reported in the distance measured in the 6-Minute Walk Test (6MWT) by 17 m (median, IQR: 0–42.5, *p* < 0.001). However, on the day of admission to the hospital, the median distance covered in the 6MWT did not differ between the prehabilitation group (median 450.0 m, IQR: 420.0–505.0) and the control group (median 450.0 m, IQR: 410.0–510.0) (*p* = 0.5). Moreover, significant improvements in muscle strength were observed in the prehabilitation group, as measured with the LUNA EMG device, with increases noted in both maximum and average muscle tension and muscle tone (Table 5). On the day of admission to the hospital, an important difference between the prehabilitation group and the control group was the maximum muscle tension (134.8 mV [75.1, 166.1 mV] vs. 83.5 mV [62.0, 98.4 mV], *p* = 0.006), while the minimum tension, average tension, and muscle tone did not differ significantly from each other.

In the prehabilitation group, a negative correlation was found between the maximum muscle tension and the duration of hospitalization after surgery, indicating that patients with higher maximum muscle tension measured prior to surgery had shorter hospital stays (R = −0.35, *p* = 0.039). However, no significant correlation was observed between the results of the 6MWT performed before surgery and on the day of discharge from the hospital (R = 0.016, *p* = 0.93).

No differences in VO2max measurements were reported between the study and control groups on the day before surgery (median 27.8 vs. 27.3 mL/kg^−1^ min^−1^, *p* = 0.6). Additionally, in the prehabilitation group, there was no significant difference in VO2max measured before and at the end of the prehabilitation intervention.

### 3.3. Changes in Laboratory Test Results

Despite recommendations for iron supplementation for patients with hemoglobin levels below 11.5 g/dL, no changes in hemoglobin levels or red blood counts were observed between the beginning of the prehabilitation program and the day of admission to the hospital. However, a negative correlation was observed in the prehabilitation group, indicating that lower hemoglobin concentrations were associated with longer hospital stays after surgery (R = −0.34, *p* = 0.045). Similarly, there were no observed differences in total protein and albumin concentrations, despite recommendations for a high-protein diet (1.5–2.0 g of protein per kilogram of body weight) and protein supplementation. Notably, the concentration of vitamin D increased significantly during the prehabilitation period, with a median change of 9.55 ng/mL (*p* < 0.001).

### 3.4. Assessment of Nutritional Status, Anxiety/Depression Symptoms, Frailty, and Quality of Life

Table 6 presents the characteristics of the patients in terms of nutritional status assessed by the MNA and MUST scales, frailty score measured by the G8 scale, anxiety and depression scores evaluated using the HADS scale, and quality of life assessed through the EORTC-QLQ-C30. No significant differences were noted between the prehabilitation group and the control group.

Additionally, within the prehabilitation group, an additional evaluation of quality of life and assessment of anxiety and depression were conducted, comparing the beginning and end of the prehabilitation program (Table 7). The analysis revealed that prehabilitation had no major impact on the change in the quality of life or the severity of anxiety and depression symptoms during the prehabilitation program.

### 3.5. Complications

In the prehabilitation group, there were more patients without postoperative complications compared to the control group (47.2% vs. 20.6% of patients without postoperative complications according to the Clavien–Dindo classification, *p* = 0.02). However, no significant differences were noted in the frequency of intensive care unit (ICU) admissions, need for reoperation, the dehiscence of intestinal anastomosis, the occurrence of eventration, cardiovascular/pulmonary complications, or hospital readmission. Two prehabilitated patients required reoperation due to suspected eventration or abdominal bleeding compared to four patients from the control group (*p* = 0.4). Patients in the prehabilitation group also required significantly fewer postoperative transfusions of packed red blood cells compared to the control group (14% vs. 47%, *p* = 0.002). Notably, patients in the prehabilitation group experienced shorter postoperative hospital stays compared to those in the control group. The median day of discharge from the hospital after surgery was 5.0 days (IQR: 4.0–6.2) in the prehabilitation group, compared to 7.0 days (IQR: 6.0–10.0) in the control group (*p* < 0.001). Postoperative outcomes of prehabilitation patients compared to the control group are presented in Table 8.

A negative correlation was reported between the maximum muscle tension measured on the day of admission to the hospital and the number of days staying in the hospital after surgery in both the entire patient group and the prehabilitation group. Interestingly, the correlation was slightly stronger in the prehabilitation group compared to the entire patient group (R = −0.32, *p* = 0.008 vs. R = −0.35, *p* = 0.039). However, no relationship was found between the value of the 6MWT on the day of admission to the hospital and that on the day of discharge from the hospital after surgery. The odds of postoperative complications among prehabilitation patients were 0.20 times (80%) lower than in the control group (Table 9).

## 4. Discussion

Our study revealed improved surgical outcomes among patients undergoing a prehabilitation program before cytoreductive surgery for advanced ovarian cancer or suspected ovarian tumors. We observed reductions in postoperative length of stay and the incidence of complications classified according to the Clavien–Dindo system in the prehabilitation group. We demonstrated that the odds of postoperative complications among prehabilitation patients were 0.20 times (80%) lower than in the control group. Additionally, a decreased need for blood transfusions in the postoperative period was observed among patients undergoing prehabilitation. However, the intervention did not affect the frequency of other major complications such as reoperations or readmissions.

This study presents the results of the effectiveness of a prehabilitation program among patients undergoing cytoreductive surgery for advanced ovarian cancer or suspected ovarian tumors. The findings suggest that prehabilitation is a safe program with no contraindications for implementation into routine practice. Our results demonstrate that prehabilitation is associated with shorter hospital stays after surgery and fewer postoperative complications.

Existing research also supports the potential benefits of prehabilitation programs for patients undergoing surgical procedures [5,10,11,12,13,14,23,24,25,26,27]. In our study, we found a significant reduction in hospitalization time after surgery in the prehabilitation group, consistent with findings from numerous other studies [10,11,12,13,27]. We also reported an overall reduction in the number of complications according to the Clavien–Dindo classification in the prehabilitation group. Studies assessing the impact of prehabilitation on the occurrence of postoperative complications assessed the length of hospital stay, ICU hospitalization, and the need for reoperation or rehospitalization. Wooten et al. showed a shorter length of hospital stay among prehabilitated patients and a decreased incidence of any complications, although they did not demonstrate a decrease in the incidence of serious complications [10]. In the study presented by Koh et al., the duration of hospitalization in the prehabilitation group was shorter, but the morbidity rate was similar to the control group [12]. The study presented by Bojessen et al. revealed that prehabilitation intervention was associated with an absolute risk reduction in severe complications such as a prolongated hospital stay, ICU hospitalization, or death. In our study, there were no serious complications such as death, but we did not observe any differences in individual serious complications, such as the frequency of reoperations or hospitalization in the ICU. Importantly, prehabilitation has also been associated with decreased hospital costs [12,13].

Conversely, a review by Molenaar et al. highlights studies [28,29,30] that did not find a positive effect of prehabilitation on postoperative complications. However, it is important to consider certain factors. In all three studies, the majority of patients in the prehabilitation group underwent minimally invasive surgery [28,29,30].

In the second study, patients with metastatic cancer and comorbidities contraindicating exercise were excluded [29]. Additionally, it is worth noting that these studies focused on patients with colorectal cancer, whereas our intervention targets patients with ovarian cancer undergoing open cytoreductive surgery.

It seems that the concept of prehabilitation aims to identify patients who may benefit the most from the intervention. This subset typically includes frail, elderly patients with multiple comorbidities undergoing radical cytoreductive open abdominal surgery. It has been shown that high-risk patients benefit most from prehabilitation interventions [5,10,14,23,27].

While the components of prehabilitation programs are well known, a precise prehabilitation protocol, particularly concerning the type of physical exercise, has yet to be developed [5]. Furthermore, there is a lack of clear tools for objectively assessing patients’ physical capacity.

In prehabilitation studies, physical fitness is often evaluated using the 6MWT [10,11,12,13]. Additionally, other assessment tools such as grip strength, gait speed, 30 s chair rise repetitions, functional reach [12], timed up and go test (TUG) [14], or cardiopulmonary exercise test (CPET) on a cycle ergometer [23] have been employed to evaluate physical parameters.

In our study, we measured muscle tension with the LUNA EMG device. To our knowledge, this is the first study to employ an objective assessment of muscle tone for baseline evaluation and to monitor the progress of physical preparation in a prehabilitation program. Our hypothesis posited that patients who adequately prepared for surgery would exhibit higher muscle tension after the prehabilitation period. That hypothesis was confirmed, and we found an increase in muscle strength, thereby concluding that LUNA EMG might be useful to monitor adherence to the program. The LUNA EMG examination revealed greater muscle strength resulting from the patients’ preparation in the prehabilitation program, which could be associated with better postoperative results in this group of patients.

Both resistance interval training and cardio training are known to enhance VO2max and muscle endurance, resulting in improved physical performance [31,32]. Additionally, it is well established that higher levels of physical fitness correlate with a reduced risk of postoperative complications [33].

Adherence to the second pillar of the prehabilitation program, which emphasizes a high-protein diet and protein supplementation, was strongly encouraged among patients. Malnutrition has been identified as a risk factor for postoperative complications [4,7,14,26,32,34], with the European Society for Clinical Nutrition and Metabolism (ESPEN) guidelines underscoring the link between nutritional status and the frequency and severity of such complications [35]. Studies have demonstrated that appropriate preoperative nutritional intervention can reduce postoperative complications by up to 20% among malnourished patients undergoing abdominal surgery [32]. Additionally, research by Manasek et al. found that implementing a high-protein diet before and after colon cancer surgery led to significant reductions in postoperative complications, including wound dehiscence, infections, intestinal anastomosis dehiscence, rehospitalization, and shortened hospital stays [36]. Moreover, combining physical training with protein supplementation has been shown to enhance the synthesis of new muscle fibers [27]. Although our prehabilitation program included a nutritional intervention with increased protein intake, we did not observe significant differences in total protein and albumin concentrations among patients in the prehabilitation group. It is possible that the relatively short duration of the prehabilitation period was insufficient to produce noticeable changes in laboratory parameters indicative of nutritional status.

In patients with anemia, despite the administration of iron supplementation, we did not observe a significant increase in hemoglobin concentration either. It is crucial to emphasize that due to the home-based prehabilitation program, oral iron supplementation was recommended. The lack of increase in hemoglobin concentration despite oral supplementation with iron preparations may be due to the route of drug administration. If feasible, intravenous iron administration should be considered in patients with preoperative anemia. However, it is noteworthy that individuals in this subgroup exhibited a notably lower frequency of blood transfusions compared to the control group, suggesting a potential benefit from the intervention.

Furthermore, we prioritized addressing vitamin D deficiencies, and our prehabilitation program resulted in a significant increase in vitamin D concentration among patients. This result allows us to conclude that the patients followed the supplementation recommendations. Research by Balci et al. highlighted a correlation between vitamin D levels and the frequency of infectious complications and surgical site infections [37]. Similarly, other studies have indicated a link between vitamin D deficiency and a higher risk of postoperative complications [22,38]. Moreover, maintaining adequate levels of vitamin D appears to positively impact the body’s response to chemotherapy [39]. This underscores the significance of prehabilitation not only in enhancing functional capacity during surgery but also in preparing patients for subsequent stages of oncological treatment.

In our study, no significant differences were reported in terms of the quality of life or the severity of anxiety and depression during the prehabilitation intervention, nor were there any disparities between the prehabilitation group and the control group. However, it is important to note that other studies have shown associations between prehabilitation and improved quality of life [12,40,41]. The lack of improvement in quality of life in our study may be attributed to the relatively short preparation time for the procedure and the absence of routine, consistent psychological support for all patients, irrespective of their individual needs. Nevertheless, it is worth noting that in the preoperative period, which is a difficult time for oncology patients, the quality of life and the severity of anxiety and depression did not deteriorate.

There are several limitations to our study. Firstly, ensuring the quality and consistency of physical exercises and adherence to the recommended diet was challenging, despite efforts to provide remote support and assistance. Secondly, the target population for prehabilitation, which typically includes seriously ill, elderly individuals with multiple limiting factors, faces various challenges. For instance, increasing ascites, often accompanied by dyspnea, can significantly hinder engagement in physical activity. Although there are no specific guidelines for the type of recommended physical exercise, it is crucial to tailor the regimen to individual patients, especially those with numerous comorbidities.

## 5. Conclusions

The prehabilitation program proposed in this study improves muscle strength and physical performance, which reduces the number of postoperative complications and shortens the length of hospital stays. Testing muscle tension with the LUNA EMG device can be a useful tool for monitoring the progress of physical preparation.

## Figures and Tables

**Figure 1 cancers-16-02493-f001:**
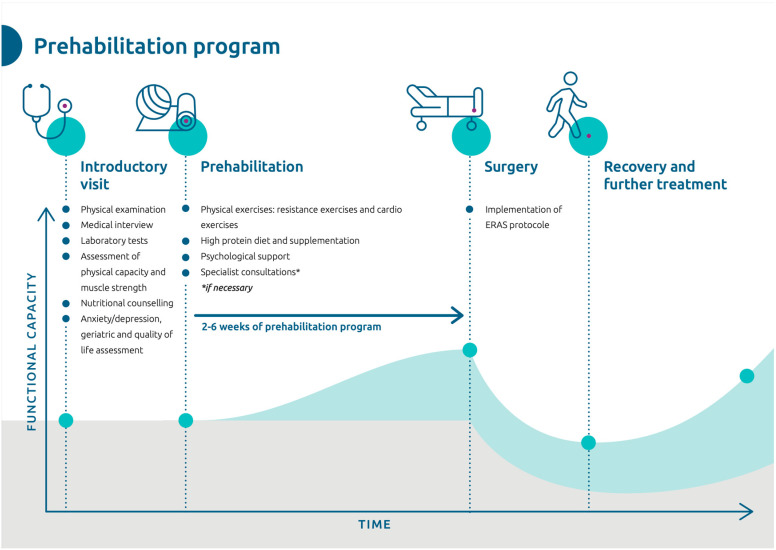
Scheme showing the prehabilitation program.

**Table 1 cancers-16-02493-t001:** General characteristics of the prehabilitation group and the control group before starting the prehabilitation program.

Characteristics	Prehabilitation Group (*n* Total = 36)	Control Group (*n* Total = 34)	*p*-Value
Age (years), median [IQR]	56.6 (49.8, 68.0)	51.0 (47.2, 64.0)	0.2
BMI (kg/m^2^), median [IQR]	26.6 (23.1, 29.1)	27.3 (25.4, 29.9)	0.3
ECOG 0, *n*/*n* total (%)	33/36 (92)	31/34 (91)	1.0
Diabetes, *n*/*n* total (%)	7/36 (19)	2/34 (5.9)	0.2
Hypertension, *n*/*n* total (%)	15/36 (42)	14/34 (41)	1.0
Ischemic heart disease, *n*/*n* total (%)	5/36 (14)	3/34 (8.8)	0.7
Smoking, *n*/*n* total (%)	8/36 (22)	6/34 (18)	0.6
Retired, *n*/*n* total (%)	10/36 (28)	13/34 (38)	0.5
CA-125 (U/mL), median (IQR)	175.0 (34.8, 950.3)	170.0 (30.0, 535.0)	0.4
Albumin concentration (g/dL), median (IQR)	4.5 (4.2, 4.8)	4.4 (4.1, 4.6)	0.3
Total protein concentration (g/dL), median (IQR)	7.3 (7.0, 7.5)	7.4 (6.8, 7.7)	0.7
Creatinine (mg/dL), median (IQR)	0.7 (0.6, 0.8)	0.8 (0.7, 0.9)	0.1

Abbreviations: IQR—interquartile range, ECOG—Eastern Cooperative Oncology Group.

**Table 2 cancers-16-02493-t002:** Surgical characteristics of the prehabilitation group and the control group.

Characteristics	Prehabilitation Group (*n* Total = 36)	Control Group (*n* Total = 34)	*p*-Value
Duration of surgery, [min], average (SD)	215.0 (108)	260.0 (93)	0.08
Type of surgery			0.13
Primary surgery, *n*/*n* total (%)	32 (88.9)	26 (76.5)	
Interval debulking surgery, *n*/*n* total (%)	4 (11.1)	4 (11.8)	
Secondary cytoreduction, *n*/*n* total (%)	0	4 (11.8)	
Residual disease			0.61
R0, *n*/*n* total (%)	33 (92)	32 (94)	
R1, *n*/*n* total (%)	1 (3)	2 (6)	
R2, *n n*/total (%)	2 (6)	0	
Aletti complexity score, median (IQR)	4.0 (2.0, 5.0)	4.0 (3.0, 5.0)	0.8
PCI, median (IQR)	9.0 (3.8, 16.8)	7.0 (2.5, 11.0)	0.1

Abbreviations: IQR—interquartile range, SD—standard deviation, PCI—peritoneal cancer index.

**Table 3 cancers-16-02493-t003:** Surgical characteristics with performed procedures during the surgery in the prehabilitation group and the control group.

Characteristics	Prehabilitation Group (*n* Total = 36)	Control Group (*n* Total = 34)	*p*-Value
Hysterectomy with bilateral Salpingo-oophorectomy, *n*/*n* total (%)	35/36 (97.2)	29/34 (85.3)	0.1
Fertility preserving surgery, *n*/*n* total (%)	1/36 (2.8)	1/34 (2.9)	1.0
Omentectomy, *n*/*n* total (%)	31/36 (86.1)	27/34 (79.4)	0.5
Appendectomy, *n*/*n* total (%)	23/36 (63.9)	18/34 (52.9)	0.5
Round ligament of the liver resection, *n*/*n* total (%)	25/36 (69.4)	20/34 (58.8)	0.5
Pelvic peritonectomy, *n*/*n* total (%)	12/36 (33.3)	8/34 (23.5)	0.4
Diaphragmatic stripping, *n*/*n* total (%)	7/36 (19.4)	5/34 (14.7)	0.8
Lesser omentum resection, *n*/*n* total (%)	1/36 (2.8)	1/34 (2.9)	1.0
Pelvic lymphadenectomy, *n*/*n* total (%)	5/36 (13.9)	10/34 (29.4)	0.1
Paraaortic lymphadenectomy, *n*/*n* total (%)	6/36(16.7)	5/34 (14.7)	1.0
Splenectomy, *n*/*n* total (%)	3/36 (8.3)	0/34	0.2
Colorectal resection, *n*/*n* total (%)	5/36 (13.9)	8/34 (23.5)	0.4
Intestinal resection, *n*/*n* total (%)	0/36	2/34 (5.9)	0.2
Partial liver resection, *n*/*n* total (%)	1/36 (2.8)	1/34 (2.9)	1.0
Partial pancreatic resection, *n*/*n* total (%)	1/36 (2.8)	0/34	1.0
Partial gastrectomy, *n*/*n* total (%)	1/36 (2.8)	0/34	1.0
Intestinal anastomosis, *n*/*n* total (%)	3/36 (8.3)	9/34 (26.5)	0.059
Stoma, *n*/*n* total (%)	2/36 (5.6)	1/34 (2.9)	1.0

**Table 4 cancers-16-02493-t004:** Histopathological characteristics of the prehabilitation group and the control group.

Characteristics	Prehabilitation Group (*n* Total = 36)	Control Group (*n* Total = 34)	*p*-Value
Ovarian cancer confirmed, *n*/*n* total (%)	28/36 (77.8)	28 (82.4)	0.8
Adenocarcinoma serous, *n*/*n* total (%)	14/36 (50)	12 (42.9)	0.8
Adenocarcinoma mucinous, *n*/*n* total (%)	2/36 (7.1)	2 (7.1)	1.0
Adenocarcinoma endometrioid, *n*/*n* total (%)	1/36 (3.6)	4 (14.3)	0.2
Clear cell adenocarcinoma, *n*/*n* total (%)	2/36 (7.1)	4 (14.3)	0.4
Neuroendocrine adenocarcinoma, *n*/*n* total (%)	1/36 (3.6)	0 (0)	1.0
Sarcoma, *n*/*n* total (%)	1/36 (3.6)	1 (3.6)	1.0
Undifferentiated carcinoma, *n*/*n* total (%)	1/36 (3.6)	1 (3.6)	1.0
Borderline cancer, *n*/*n* total (%)	4/36 (14.3)	1 (3.6)	0.4
Other, *n*/*n* total (%)	2/36 (7.1)	3 (10.7)	0.7
Low-grade carcinoma, *n*/*n* total (%)	5/36 (17.9)	9 (32.1)	0.2
High-grade carcinoma, *n*/*n* total (%)	17/36 (60.7)	16 (57.1)	1.0
FIGO stages, *n*/*n* total (%)			1.0
FIGO 1 stage	8/36 (22.2)	8 (23.5)	
FIGO 2 stage	1/36 (2.8)	1 (2.9)	
FIGO 3 stage	19/36 (52.8)	17 (50)	
FIGO 4 stage	1/36 (2.8)	1 (2.9)	

**Table 5 cancers-16-02493-t005:** Changes in values in LUNA EMG measurements during the prehabilitation period.

Characteristics	Introductory Visit	Day of Admission to Hospital	Change	*p*-Value
Maximum muscle tension [mV], median [IQR]	83.7 (51.2, 143.3)	134.8 (75.1, 166.1)	20.4 (4.1–50.1) *	<0.001
Average muscle tension [mV], median [IQR]	14.3 (9.8, 26.1)	21.4 (12.6, 36.2)	4.3 (0.3–13.3) *	<0.001
Muscle tone [mV], median [IQR]	2.0 (1.2, 2.8)	2.4 (1.5, 3.3)	0.3 (0–1.0) *	0.004

Abbreviations: IQR—interquartile range. * Change was defined as the difference in the parameter’s value between the end and the beginning of the prehabilitation program. This involved maximum muscle tension, average muscle tension, and muscle tone, which were measured as peak tension during muscle contraction, average muscle tension during muscle activity, and constant muscle tension, respectively.

**Table 6 cancers-16-02493-t006:** Characteristics of the prehabilitation group and control group in terms of nutritional status, frailty, anxiety and depression score, and quality of life on the day of admission to the hospital.

Characteristics	Prehabilitation Group (*n* Total = 36)	Control Group (*n* Total = 34)	*p*-Value
MNA			0.8
Normal nutritional status, *n*/*n* total (%)	21/36 (58)	16/34 (53)	
At risk of malnutrition, *n*/*n* total (%)	13/36 (36)	13/34 (43)	
Malnourished, *n*/*n* total (%)	2/36 (5.6)	1/34 (3.3)	
MUST			0.9
0, *n*/*n* total (%)	32/36 (89)	29/34 (85)	
1, *n*/*n* total (%)	4/36 (11)	4/34 (12)	
2, *n*/*n* total (%)	0/36 (0)	1/34 (2.9)	
G8, median [IQR]	14.0 (12.0, 15.0)	14.0 (12.5, 16.0)	0.4
HADS Anxiety			0.8
Mild, *n*/*n* total (%)	8/36 (22)	7/34 (21)	
Moderate, *n*/*n* total (%)	7/36 (19)	5/34 (15)	
Normal, *n*/*n* total (%)	20/36 (56)	22/34 (65)	
Severe, *n*/*n* total (%)	1/36 (2.8)	0/34 (0)	
HADS Depression			0.7
Mild, *n*/*n* total (%)	3/36 (8.3)	5/34 (15)	
Moderate, *n*/*n* total (%)	5/36 (14)	3/34 (8.8)	
Normal, *n*/*n* total (%)	27/36 (75)	26/34 (76)	
Severe, *n*/*n* total (%)	1/36 (2.8)	0/34 (0)	
EORTC-QLQ-C30 QLQ Total, median [IQR]	84.1 (77.1, 92.2)	81.8 (72.8, 88.4)	0.2

Abbreviations: IQR—interquartile range, MNA—Mini Nutritional Assessment, MUST—Malnutrition Universal Screening Tool, G8—Geriatric 8, HADS—Hospital Anxiety and Depression Scale, EORTC—European Organization for Research and Treatment of Cancer.

**Table 7 cancers-16-02493-t007:** Changes in quality of life and severity of anxiety and depression symptoms during the prehabilitation program.

Characteristics	Beginning of Prehabilitation	End of Prehabilitation	*p*-Value
EORTC-QLQ-C30 QLQ Total, median [IQR]	82.1 (72.8, 88.6)	84.1 (77.1, 92.2)	0.2
HADS Anxiety			0.7
Mild, *n*/*n* total (%)	4/36 (11)	8/34 (22)	
Moderate, *n*/*n* total (%)	10/36 (28)	7/34 (19)	
Normal, *n*/*n* total (%)	21/36 (58)	20/34 (56)	
Severe, *n*/*n* total (%)	1/36 (2.8)	1/34 (2.8)	
HADS Depression			0.7
Mild, *n*/*n* total (%)	4/36 (11)	8/34 (22)	
Moderate, *n*/*n* total (%)	10/36 (28)	7/34 (19)	
Normal, *n*/*n* total (%)	21/36 (58)	20/34 (56)	
Severe, *n*/*n* total (%)	1/36 (2.8)	1/34 (2.8)	

Abbreviations: IQR—interquartile range, HADS—Hospital Anxiety and Depression Scale, EORTC—European Organization for Research and Treatment of Cancer.

**Table 8 cancers-16-02493-t008:** Postoperative outcomes in prehabilitation group vs. control group.

Characteristics	Prehabilitation Group (*n* Total = 36)	Control Group (*n* Total = 34)	*p*-Value
Intensive care unit hospitalization, *n*/*n* total (%)	2/36 (6)	3/34 (9)	0.7
Reoperation, *n*/*n* total (%)	2/36 (6)	4/34 (12)	0.4
Dehiscence of intestinal anastomosis, *n*/*n* total (%)	1/36 (3)	2/34 (6)	0.5
Wound dehiscence with eventration, *n*/*n* total (%)	1/36 (3)	0/34 (0)	1.0
Cardiovascular complications, *n*/*n* total (%)	1/36 (3)	0/34 (0)	1.0
Pulmonary complications, *n*/*n* total (%)	0/36 (0)	0/34 (0)	1.0
Lymphocele, *n*/*n* total (%)	0/36 (0)	0/34 (0)	1.0
Need for blood transfusion, *n*/*n* total (%)	5/36 (14)	16/34 (47)	0.002
Day of discharge from the hospital, median (IQR)	5.0 (4.0, 6.2)	7.0 (6.0, 10.0)	<0.001
Readmission within 30 days, *n*/*n* total (%)	1/36 (3)	4/34 (12)	0.2
Without any complications according to Clavien-Dindo classification, *n*/*n* total (%)	17/36 (47)	7/34 (21)	0.02

Abbreviations: IQR—interquartile range.

**Table 9 cancers-16-02493-t009:** Univariate and multivariate logistic regression analyses considering postoperative complications.

Risk Factor	Univariate Logistic Regression		Multiple Logistic Regression	
Characteristic	OR (95% CI)	*p*-value	OR (95% CI)	*p*-value
Age (years)	1.00 (0.96, 1.04)	0.92		
BMI (kg/m^2^)	1.04 (0.93, 1.17)	0.45		
Prehabilitation				
No	Ref.		Ref.	
Yes	0.29 (0.10–0.81)	0.018	0.20 (0.04–0.80)	0.031
6MWT ^1^ (m)	1.00 (0.99–1.00)	0.32		
FIGO				
1–2	Ref.			
3–4	2.49 (0.75–8.40)	0.13	1.87 (0.43–8.53)	0.4
Total protein concentration ^1^ (g/dL)	1.62 (0.90–4.11)	0.12	3.15 (1.12–23.2)	0.2
Albumin concentration ^1^ (g/dL)	0.37 (0.09–1.18)	0.094	0.45 (0.07–2.32)	0.4
Aletti complexity score	1.52 (1.09–2.28)	0.011	1.53 (1.00–2.62)	0.083

Abbreviations: OR—odds ratio, CI—confidence interval; ^1^ measured on the day of admission to hospital (the day before surgery).

## Data Availability

Data are contained within the article.

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
