# Peer review of "LUNA EMG as a Marker of Adherence to Prehabilitation Programs and Its Effect on Postoperative Outcomes among Patients Undergoing Cytoreductive Surgery for Ovarian Cancer and Suspected Ovarian Tumors"

_cancers, 2024, doi:10.3390/cancers16142493_

Round 1

Reviewer 1 Report (Previous Reviewer 1)

Comments and Suggestions for Authors

Dear Authors,

Thank you for going through my comments.

Best of luck with your future research!

Reviewer 2 Report (Previous Reviewer 2)

Comments and Suggestions for Authors

Authors adequately addressed the reviewers' issues

This manuscript is a resubmission of an earlier submission. The following is a list of the peer review reports and author responses from that submission.

Round 1

Reviewer 1 Report

Comments and Suggestions for Authors

Dear Authors,

I was glad to come across your paper. I believe that such interventions in cancer patients are highly needed and important and should be included in the general routine. Unfortunately, few centers actually emphasize it.

Please find my suggestiions in the attached file.

Also: Figure 1 error: Implementation of ERAS protocol*E*

I look forward to your revised work!

Comments on the Quality of English Language

The manuscript has issues with prepositions and articles, verbs, plurals etc. I marked as many as I could. Thorough language editing should be done to ensure the readability of the paper.

Reviewer 2 Report

Comments and Suggestions for Authors

The authors present the results of their prospective, randomized study of the impact of a prehabilitation program on the outcomes for women with presumed ovarian cancer undergoing definitive surgery. In addition to “traditional” means of assessment (e.g., lab tests, standardized questionnaires), they used a device, LUNA EMG, to assess muscle strength and tone.

According to their results, “Within the prehabilitation group, we observed a significant increase in the 6-Minute Walk Test distance by 17 meters (median, IQR: 0-42.5, p < 0.001) and a significant increase in muscle strength measured with the LUNA EMG. In comparison to the control group, the prehabilitation group showed fewer complications according to the Clavien-Dindo classification (47.2% vs 20.6%, p = 0.02) and shorter post-operative hospital stays (Median 5.0 days [IQR: 4.0-6.2] vs 7.0 days [IQR: 6.0-10.0], p < 0.001).”

The study is interesting in concept and contemporary.  However, I have a number of concerns:

1)      The randomization scheme, “every second patient referring to the Clinic was selected to prehabilitation group”, is far less than ideal with substantial potential for bias. For example, the chosen method fails to ensure absence of a priori knowledge of the group assignment (blinding) and creates the potential for selection bias.

2)      The statistical plan lacks a power analysis, crippling the reviewer’s and reader’s ability to assess the results of the study. Overall, many of their reported outcomes fail to show a difference.  The authors fail to provide any basis for assessing the potential for a Type II error. In addition, it does not appear an adequate multivariate analysis was conducted/presented.

3)      The actual selection and distribution of the patients is unclear. According to the manuscript, 70 patients with known or suspected ovarian cancer were enrolled, 36 in the prehabilitation group and 34 in the control group. Of the entire group, 56 patients had confirmed ovarian cancer, but it unclear how many in each group had cancer. If the study was an assessment of “LUNA EMG as a Marker of Adherence to Prehabilitation Program and its Effect on Postoperative Outcomes among Patients Undergoing Cytoreductive Surgery for Ovarian Cancer” (underline added for emphasis), why were the 14 patients without ovarian cancer included in the analysis? Furthermore, according to the manuscript, “Patients with advanced disease, for whom cytoreductive surgery was not feasible due to disease progression observed during laparoscopy, were excluded from the study. However, they still underwent the prehabilitation program after neoadjuvant chemotherapy and reassessment”. The meaning of this statement on patient selection and inclusion is unclear.

4)      There are no data on the surgical procedures except for average operating time.  Therefore, there is no way to assess whether the surgical procedures were comparable for the groups.

5)      The discussion of the complications is inadequate.  Although the authors report that there was a lower incidence of postoperative complications in the prehabilitation group, no data were provided for the post-operative complications suffered by each group except to indicate there were “no significant differences were noted in the frequency of intensive care unit admissions, need for reoperation, occurrence of eventration, or hospital readmission.”

6)      Overall, the discussion was inadequate and failed to explain their findings, negative and positive.

Comments on the Quality of English Language

Needs moderate editing

Reviewer 3 Report

Comments and Suggestions for Authors

This study presents results on the effectiveness of prehabilitation program in patients undergoing cytoreductive surgery for advanced ovarian cancer

In this study, the authors used the LUNA EMG device to assess the effectiveness of a prehabilitation program in patients undergoing cytoreductive surgery for advanced ovarian cancer. They showed a significant improvement in muscle strength as  measured by the LUNA EMG device, with an increase in both maximum and average  muscle tension and muscle tone. This was associated with reduced postoperative hospital stay and fewer postoperative complications.

This is an interesting study. However there are some comments to this manuscript.

In Table 1 it should be clear whether the values in the Prehabilitation group are before or at the completion (admission day) of the Prehabilitation program. In addition it should be clear to what the percentages refer (for example, 19% of the study group and 5.9% of the control group were diabetics?). Please give also units for Age and BMI.

Table 2. Please give a more descriptive header or add a footnote. These are changes LUNA EMG measurements after completion of the prehabilitation program, apparently values represent an increase from baseline measurements. This should be clear. Please also clarify maximum muscle tension, average muscle tension and muscle tone.

Tables 3 and 4. Please also clarify n(%) for the data presented.

The sentence “… a lower incidence of postoperative complications was observed compared to the control group (47.2% vs. 20.6% of patients without postoperative complications …” (paragraph 3.5) does not make sense. Please clarify by rephrasing.

The authors state that prehabilitation is associated fewer postoperative complications but there is mention to the type of complications.

Please make sure that all abbreviations are given in full when first mentioned (see VO2, EPA, DHA and others)

The manuscript needs also some linguistic editing.

Comments on the Quality of English Language

Some minor editing is needed